# The Mitochondrial Distribution and Morphology Family 33 Gene *FgMDM33* Is Involved in Autophagy and Pathogenesis in *Fusarium graminearum*

**DOI:** 10.3390/jof10080579

**Published:** 2024-08-16

**Authors:** Wuyun Lv, Yiyi Tu, Ting Xu, You Zhang, Junjie Chen, Nan Yang, Yuchun Wang

**Affiliations:** 1College of Tea Science and Tea Culture, Zhejiang A&F University, Hangzhou 311300, China; 2022614022068@stu.zafu.edu.cn (Y.T.); 2022614022085@stu.zafu.edu.cn (T.X.); 2022614022096@stu.zafu.edu.cn (Y.Z.); 2023115011001@stu.zafu.edu.cn (J.C.); 2The People’s Government Office of Bengbu City, Bengbu 233000, China; bbszfbxzk@163.com

**Keywords:** mitochondrial fission, *MDM33*, autophagy, pathogenesis, apoptosis, *Fusarium graminearum*

## Abstract

The mitochondrial distribution and morphology family 33 gene (*MDM33*) regulates mitochondrial homeostasis by mediating the mitochondrial fission process in yeast. The wheat head blight *Fusarium graminearum* contains an FgMdm33 protein that is orthologous to *Saccharomyces cerevisiae* Mdm33, albeit its function remains unknown. We have reported here the roles of FgMdm33 in regulating fungal morphogenesis, mitochondrial morphology, autophagy, apoptosis, and fungal pathogenicity. The Δ*Fgmdm33* mutants generated through a homologous recombination strategy in this study exhibited defects in terms of mycelial growth, conidia production, and virulence. Hyphal cells lacking *FgMDM33* displayed elongated mitochondria and a dispensable respiratory-deficient growth phenotype, indicating the possible involvement of *FgMDM33* in mitochondrial fission. The Δ*Fgmdm33* mutants displayed a remarkable reduction in the proteolysis of GFP-FgAtg8, whereas the formation of autophagic bodies in the hyphal cells of mutants was recorded under the induction of mitophagy. In addition, the transcriptional expression of the apoptosis-inducing factor 1 gene (*FgAIF1*) was significantly upregulated in the Δ*Fgmdm33* mutants. Cumulatively, these results indicate that *FgMDM33* is involved in mitochondrial fission, non-selective macroautophagy, and apoptosis and that it regulates fungal growth, conidiation, and pathogenicity of the head blight pathogen.

## 1. Introduction

The mitochondria play a central role in the cellular metabolism of eukaryotic cells by acting as the powerhouses to generate ATP through the tricarboxylic acid cycle and oxidative phosphorylation [1,2,3]. To maintain their proper functional state, mitochondria undergo dynamic processes, involving fission, fusion, transport, and mitophagy [3,4]. Fusion and fission are crucial for mitochondrial homeostasis, which regulates mitochondrial morphology, distribution, and function [5]. The fusion process is beneficial for ATP generation and it maintains the mitochondrial DNA (mtDNA) levels and fidelity, whereas the fission of mitochondria influences the morphology and facilitates mitochondrial transport, mitophagy, apoptosis, and cell division [6,7,8]. 

In *Saccharomyces cerevisiae*, the mitochondrial outer membrane proteins Fzo1 and Ugo1 and inner membrane protein Mgm1 regulate mitochondrial fusion [9,10,11]. The conserved dynamin GTPase Dnm1 is responsible for mitochondrial fission, which is then recruited to the mitochondrial outer membrane by the mitochondrial adaptor proteins Fis1, Mdv1, and Caf4 [12,13,14,15]. Dimmer et al. reported a novel gene *MDM33* that encodes a predicted membrane protein through the systematic screen, and they observed that the majority of *mdm33* mutant cells harbored giant ring-like mitochondrial structures or 2–4 smaller interconnected mitochondrial rings [16,17]. In addition, the structural defects of other organelles, such as the endoplasmic reticulum (ER), vacuoles, and cytoskeletons, were not determined in Δ*mdm33* cells [17]. As a consequence, Mdm33 is specifically required for the regulation of mitochondrial morphology. The phenotype of the Δ*mdm33* mutant bears some similarities to Δ*dnm1*, Δ*mdv1*, and Δ*fis1* [17,18,19,20]. The overexpression of *MDM33* results in a high aggregation of the mitochondria, the formation of the inner membrane septa, and the disappearance of the inner membrane cristae, which together severely affects the mitochondrial morphology [17]. When combined with the location in the mitochondrial inner membrane, Mdm33 is supposedly involved in the fission of the mitochondrial inner membrane [17]. 

Yeast Mdm33 interacts with prohibitins, Phb1 and Phb2, which are the inner membrane complex that regulates the mitochondrial phospholipid homeostasis [21]. As a result, Mdm33 modulates the phospholipids’ homeostasis in the mitochondrial inner membrane by affecting the levels of phosphatidylethanolamine and cardiolipin [21]. In addition, Mdm33 interacts with the subunits of the ER–mitochondria encounter structure (ERMES), which then mediates the formation of the ER–mitochondrial contact sites and promotes the exchange of phospholipids between two organelles [22,23]. Mdm33 can regulate mitochondrial homeostasis via interaction with genes involved in the phospholipid metabolism and mitochondrial distribution and function. For cell growth, the overexpression of *MDM33* causes growth arrest [24]. However, the molecular biological function of *MDM33* in regulating mitochondrial morphology or hyphal growth has so far not been reported in filamentous fungi. 

*Fusarium graminearum* is a phytopathogen responsible for *Fusarium* head blight (FHB), a destructive fungal disease in wheat-growing areas worldwide, which leads to serious yield losses as well as decreases in the quality of cereals due to the contamination of mycotoxins [25,26,27]. This disease caused by *F. graminearum* is difficult to control because of the lack of effective wheat cultivars with high resistance and the complexity of interaction between *F. graminearum* and wheat [25,28]. As a hemibiotrophic fungus, *F. graminearum* developed infectious hyphae in subcuticular and intercellular plant tissues at the initial stages of infection, and colonized plant cells intracellularly after causing their death at later stages of infection [25,29]. In the processes, numerous genes related to the pathogenesis have been revealed [30]. Previous studies showed that some genes such as *FgNdk1* and *Fgporin* affecting the mitochondrial morphology play crucial roles in pathogenicity [31,32]. In this study, we identified *FgMDM33* (FGSG_09659) encoding a putative mitochondrial distribution and morphology family 33 protein. Targeted gene deletion mutants of *FgMDM33* exhibited a significant reduction in mycelial growth, conidia production, and virulence. The Δ*Fgmdm33* mutants exhibited a high sensitivity to H_2_O_2_ and induced the upregulation of the apoptosis-inducing factor (*AIF*). The deletion of this gene causes defects in the mitochondrial morphology and non-selective macroautophagy. Hence, in *F. graminearum*, FgMdm33 plays a crucial role in regulating fungal development, pathogenicity, and mitochondrial morphology mediated by fission, apoptosis, or autophagy. 

## 2. Materials and Methods

### 2.1. Fungal Strains and Culture Conditions

*F. granminearum* wild-type strain PH-1 [33] served as a progenitor for generating the deletion mutants of *FgMDM33*. PH-1, the Δ*Fgmdm33-3* and Δ*Fgmdm33-5* mutants, and the complemented strain cΔ*Fgmdm33* were cultured on potato dextrose agar (PDA) plates at 25 °C with 12 h light/12 h dark cycles. 

### 2.2. Gene Deletion and Complementation

A homologous recombination strategy was employed for *FgMDM33* deletion, as described elsewhere [34]. The plasmid pCAMBIA1300-neo was used for constructing the complementation vector. *FgMDM33* along with its native promoter was cloned into a pCAMBIA1300-neo vector using the CloneExpress Ⅱ One Step Cloning Kit (Nanjing Vazyme Biotech Co., Ltd., Nanjing, China). All PCR primers used in this study are listed in Appendix A. The protocols for genetic transformation in *F. graminearum* are described elsewhere [35]. Gene deletion mutants and complemented transformants were confirmed via PCR and a Southern blot analysis using the digoxigenin (DIG) high-prime DNA labeling and detection starter Kit Ⅰ (Roche, Mannheim, Germany).

### 2.3. Phenotypic Analysis

For the colony growth assay, 5 mm mycelial plugs from the edge of the 3-day-old colony were, respectively, placed on PDA, complete medium (CM, 1 g/L yeast extract, 1 g/L casamino acid, 2 g/L peptone, 10 g/L D-glucose, 5% 20× nitrate salts, 0.1% trace elements, and 1% vitamin solution), and minimal medium (MM) plates and incubated at 25 °C for 3 days. For the conidiation assay, five 5 mm mycelial plugs from each strain were inoculated into 20 mL of a carboxylmethyl cellulose (CMC) liquid medium for 4 days at 25 °C in the light. For the investigation of conidial germination, freshly harvested spores of each strain were cultured in a 2% sucrose solution and examined under a light microscope (Olympus, Tokyo, Japan) after 24 h. These experiments, with 3 replicates, were repeated thrice. 

### 2.4. Pathogenicity Assays

For Fusarium head blight symptom evaluation on wheat (the susceptible cultivar *Jimai33*), the middle spikelets of at least 10 flowering wheat heads were inoculated with mycelial plugs [33]. Symptomatic spikelets were determined after incubation for 14 days. For the infection assays on the wheat leaf, 5 mm mycelial plugs were cut from the edge of a 3-day-old colony and inoculated on at least 6 leaves of 10-day seedlings of *Jimai33*. After inoculation, the leaves were kept at 25 °C with 100% humidity in a plant growth chamber. Photographs were taken at 3 days of inoculation. The infection experiments were conducted thrice independently. 

### 2.5. Sensitivity of Mycelial Growth to Oxidative Stress

To examine the sensitivity of each strain to oxidative stress, the prepared 5 mm mycelial plugs were placed on CM and CM supplemented with 0.05% H_2_O_2_ or a 200 ppm paraquat and cultured for 3 days at 25 °C. The colony diameters were measured by the cross-intersection method. The experiments were conducted in triplicates and repeated thrice.

### 2.6. Mitochondrial Fission Assay 

Mycelial growth was measured on CM plates supplemented with glucose or lactate as the sole carbon source, which were incubated for 3 days at 25 °C and then photographed. One piece of mycelial agar from each strain was cultivated in 20 mL of a liquid CM medium at 25 °C for 24 h, and the hyphae were collected and stained with MitoTracker^TM^ Red (Invitrogen, Waltham, MA, USA), followed by observation under the Zeiss LSM 780 laser confocal microscope (Carl Zeiss, Jena, Germany). Moreover, the harvested hyphae were treated for transmission electron microscopy (TEM) assays and observed under the JEM-1230 electron microscope (JEOL, Tokyo, Japan), operating at 70 kV.

### 2.7. Western Blot Analysis

Total protein was extracted as described previously [33]. Briefly, the protein samples from each extract were fractionated by SDS-PAGE and then immunoblotted with an anti-GFP antibody (Abmart, M20004, Shanghai, China) at the recommended dilutions. HRP-AffiniPure Goat Anti-Mouse IgG (H + L) (Fdbio Science, Shanghai, China) was used as the secondary antibody at a 1:10,000–1:50,000 dilution. FDbio-Femto ECL Kit (Fdbio Science, Shanghai, China) was used to detect the chemiluminescent signals. 

### 2.8. Autophagy Assays

For the mitophagy assays, the hyphae of each strain were cultured in a liquid CM medium for 24 h and then shifted to the basal medium with glycerol (BM-G, 1.6 g/L yeast nitrogen base, 2 g/L asparagine, 1 g/L NH_4_NO_3_, and 1.5% glycerol) for 30 h, followed by 6 h of growth in the MM-N medium with 3 mM phenymethysulfonyl fluoride (PMSF). The harvested mycelium was observed under TEM, as described previously [36]. Non-selective macroautophagy was analyzed as previously described [33]. 

### 2.9. qRT-PCR Analysis

The total RNA of each sample was isolated from the harvested hyphae by using the PureLink^TM^ RNA Mini Kit (Invitrogen, Waltham, MA, USA), and cDNA was synthesized with reverse transcription by the HiScript^®^ II RT SuperMix for qPCR (+ gDNA wiper) (Nanjing Vazyme Biotech Co., Ltd., Nanjing, China). For qRT-PCR assays, ChamQ SYBR qPCR Master Mix (Nanjing Vazyme Biotech Co., Ltd., Nanjing, China) and the BIO-RAD CFX96^TM^ Real-Time System (Hercules, CA, USA) were used. The expression level of the target gene was calculated by the 2^−ΔΔCt^ method [37] using *FgACTIN* (FGSG_07335) as the internal reference. The experiment was conducted with at least 3 replicates and was repeated thrice independently. 

### 2.10. Statistical Analyses

Statistical data were analyzed with SPSS Inc. (IBM) software (https://www.ibm.com/spss, accessed on 30 July 2009) and presented as the means ± standard deviations. Statistical significance was determined by ANOVA. 

## 3. Results

### 3.1. FgMDM33 Is Involved in Vegetative Growth and Conidiogenesis

The BLASTP analysis using *S. cerevisiae* Mdm33 as a query revealed that the *F. graminearum* genome contains an Mdm33 ortholog, which was named FgMdm33 (FGSG_09659). FgMdm33 encoding a 512 aa protein possesses the remarkable She9_MDM33 domain and a transmembrane region. To investigate the biological functions of this gene in *F. graminearum*, we generated targeted gene deletion mutants of *FgMDM33* via a homologous recombination strategy. The resulting transformants with the hygromycin-resistance gene were preliminarily screened by the PCR analysis using the primer pairs listed in Appendix A, which were then confirmed by Southern blotting hybridization (Appendix A). 

Thus, we conducted a series of phenotypic analyses of the mutants by using conventional biological methods. Δ*Fgmdm33-3* and Δ*Fgmdm33-5* mutants grew significantly slower relative to the wild-type strain PH-1 on PDA, CM, and MM plates at 25 °C for 3 days (Figure 1A,B). When compared to the thick aerial mycelia of PH-1, the aerial mycelia of the Δ*Fgmdm33-3* and Δ*Fgmdm33-5* mutants were sparse and attached to the surface of the medium. The defects in the mycelial growth of the Δ*Fgmdm33-3* and Δ*Fgmdm33-5* mutants were recovered by the reintroduction of *FgMDM33* (Figure 1A,B). When culturing in the 1% MBL medium, the Δ*Fgmdm33-3* and Δ*Fgmdm33-5* mutants, respectively, produced (2.14 ± 0.23) × 10^5^ conidia/mL and (2.26 ± 0.31) × 10^5^ conidia/mL after 4 days, demonstrating a significant decrease when compared with (4.38 ± 0.18) × 10^5^ conidia/mL and (4.32 ± 0.06) × 10^5^ conidia/mL produced by PH-1 and the complemented strain cΔ*Fgmdm33*, respectively (Figure 1C). We also observed the germination of conidia produced by PH-1, Δ*Fgmdm33-3*, Δ*Fgmdm33-5*, and cΔ*Fgmdm33* under light microscopy. The microscopic examination revealed no significant difference in the germination rate of the conidia among these strains, of which >85% of the conidia could germinate in a 2% sucrose solution (Figure 1D). These results suggested that *FgMDM33* plays an important role in regulating the vegetative growth and conidia production, but not conidial germination.

### 3.2. FgMDM33 Is Required for Full Virulence in F. graminearum

To investigate the role of *FgMDM33* in *F. graminearum*, we performed a virulence analysis by inoculating flowering wheat heads with mycelial plugs from each strain. The wild-type strain PH-1 and the complemented strain cΔ*Fgmdm33* led to the development of typical scab symptoms in inoculated and nearby spikelets of wheat heads after 14 days. Rachis of the inoculated wheat heads became blighted and the spikelets dried (Figure 2A). Moreover, the grains of the infected spikelets by PH-1 and cΔ*Fgmdm33* strains became shriveled and bleached (Figure 2A). In contrast, the Δ*Fgmdm33-3* and Δ*Fgmdm33-5* mutants triggered slight scab symptoms only in the point-inoculated spikelet, but not in any nearby spikelets. Spikelets above the inoculation point remained green, and the rachis turned brown only at the inoculated point. In addition, only the grain at the point-inoculated spikelet became shriveled, and the other grains from nearby spikelets remained plump (Figure 2A). To further testify the attenuated pathogenicity of the Δ*Fgmdm33-3* and Δ*Fgmdm33-5* mutants, we inoculated the 10-day-old wheat leaves with mycelial plugs. After 3 days of inoculation, the whole leaves inoculated by PH-1 and cΔ*Fgmdm33* were observed to be covered with mycelia and had decayed, while the leaves inoculated with the Δ*Fgmdm33-3* and Δ*Fgmdm33-5* mutants were only partially covered with mycelia and had decayed only around the inoculation sites (Figure 2B). These results indicated that *FgMDM33* is required for full virulence in *F. graminearum*. As the deletion of the gene led to the development of severe mycelial growth defects, the attenuated virulence of the Δ*Fgmdm33-3* and Δ*Fgmdm33-5* mutants was likely to be the result of growth defects.

### 3.3. FgMDM33 May Be Involved in Mitochondrial Fission

In *S. cerevisiae*, Mdm33 controls mitochondrial morphology by regulating the fission of the mitochondrial inner membrane. The phenotype of the Δ*mdm33* mutant displays no mtDNA loss and no acquisition of a respiratory-deficient growth, which bears some similarities to the deletion mutants of *DNM1*, *MDV1,* and *FIS1* genes, which are known as the critical regulators affecting mitochondrial fission [17]. To examine whether *FgMDM33* is involved in mitochondrial fission, we determined the response of the mutants to a non-fermentative carbon source. In consistence with the wild-type strain PH-1 and the complemented strain cΔ*Fgmdm33*, the Δ*Fgmdm33-3* and Δ*Fgmdm33-5* mutants grew well on CM plates using lactate as the sole carbon source (Figure 3A), which implies that the mutants did not acquire a respiratory-deficient growth phenotype. A similar phenotype suggested that *FgMDM33* may also be involved in mitochondrial fission.

To further determine the role of *FgMDM33* in mitochondrial fission, we observed the mitochondrial morphology in hyphal cells of each strain by staining mitochondria with the MitoTracker^TM^ Red dye. Under laser scanning confocal microscopy, the mitochondria in the hyphal cells of PH-1 and cΔ*Fgmdm33* were observed to be punctuated and short rod-shaped, while the mitochondria of the Δ*Fgmdm33-3* and Δ*Fgmdm33-5* mutants presented an elongated shape (Figure 3B). Moreover, we examined the morphology of the mitochondria in different strains by TEM after culturing in the CM medium for 24 h. The elongated mitochondria were observed in the Δ*Fgmdm33-5* mutant (Figure 3C). The elongated mitochondrial morphology of the Δ*Fgmdm33-3* and Δ*Fgmdm33-5* mutants reminds us that *FgMDM33* assists in maintaining the mitochondrial morphology by regulating the mitochondrial fission in *F. graminearum*.

### 3.4. Loss of FgMDM33 Caused Partial Defects in Autophagy

Mitochondria with abnormal morphology can be degraded by mitophagy [38], and the deletion of *FgMDM33* can result in defects of the mitochondrial morphology. Thus, we speculated that *FgMDM33* may be involved in mitophagy. We previously reported that FgAtg20 is involved in multiple autophagic processes but not required for mitophagy [33]. Thus, the Δ*Fgatg20* mutant was used as the negative control in the mitophagy analysis. Each strain was cultured in the glycerol medium for 30 h after culturing in the CM medium for 24 h, and then shifted to the MM-N medium for another 12 h in the presence of 2 mM phenylmethanesulfonylfluoride fluoride (PMSF). Autophagic bodies in vacuoles were observed in the wild-type strain PH-1, the Δ*Fgmdm33-5* mutant, and the complemented strain cΔ*Fgmdm33* by TEM, as in the Δ*Fgatg20* mutant (Figure 4A). This result suggested that *FgMDM33* was not involved in mitophagy in *F. graminearum*. Then, we attempted to determine whether *FgMDM33* is involved in the non-selective macroautophagy. The previously constructed GFP-FgAtg8 vector was, respectively, transformed into PH-1 and the Δ*Fgmdm33-5* mutant [33]. In the nutrient-rich CM medium, both GFP-FgAtg8 and a free GFP band could be detected in PH-1 and the Δ*Fgmdm33-5* mutant (Figure 4B). When being induced in the MM-N medium for 16 h, the level of full-length GFP-FgAtg8 decreased and an increasingly stronger free GFP band was detected in both strains. Compared to the nearly invisible GFP-FgAtg8 band in PH-1, the GFP-FgAtg8 band in the Δ*Fgmdm33-5* mutant remained strong under the condition of nitrogen starvation for 16 h (Figure 4B). The GFP-FgAtg8 proteolysis assay confirmed that the non-selective macroautophagy was partially impaired in the Δ*Fgmdm33-5* mutant. Cumulatively, *FgMDM33* is important for non-selective macroautophagy, but not for mitophagy in *F. graminearum*.

### 3.5. FgMDM33 Is Involved in the Apoptosis of F. graminearum

We analyzed the responses of each strain to several stresses and found that the Δ*Fgmdm33-3* and Δ*Fgmdm33-5* mutants displayed a high sensitivity to oxidative stress. When compared to the wild-type strain PH-1 and the complemented strain cΔ*Fgmdm33* with the mycelial growth inhibition of 59.28 ± 6.18% and 66.41 ± 0.34%, respectively, Δ*Fgmdm33-3* and Δ*Fgmdm33-5* presented 96.93 ± 2.39% and 99.29 ± 0.36% mycelial growth inhibition, respectively, and were more sensitive to 0.05% H_2_O_2_ (Figure 5A,B). Paraquat is also regarded as the agent causing oxidative stress [39]; hence, the sensitivity of each strain to paraquat was also tested. However, the sensitivity to 200 ppm paraquat displayed no significant difference among the tested strains. qRT-PCR was further conducted to investigate the expression levels of peroxidase genes involved in the regulation of oxidative stresses. Several catalase genes, *FCA1*, *FCA2,* and *FCA4*, in response to oxidative stress were found to be significantly upregulated in the Δ*Fgmdm33-3* and Δ*Fgmdm33-5* mutants when compared to PH-1 and cΔ*Fgmdm33*, although the expression level of *SOD1,* which is responsible for the detoxification of paraquat, displayed no significant difference among all strains (Figure 5C). This result suggested that the increased sensitivity of the Δ*Fgmdm33-3* and Δ*Fgmdm33-5* mutants to H_2_O_2_ was not attributable to the deficiency of the expression of catalase genes. It has been reported that H_2_O_2_ can induce apoptosis [40]; therefore, we speculated that the enhanced sensitivity of the Δ*Fgmdm33-3* and Δ*Fgmdm33-5* mutants to H_2_O_2_ may be attributable to the activation of apoptotic cell death after the deletion of *FgMDM33*. AIF is the main mediator of caspase-independent apoptosis-like cell death, and its expression is upregulated during the process of apoptosis [41]. The qRT-PCR analysis in the present study showed that the expression of *FgAIF1* (FGSG_02433), the AIF in *F. graminearum* (Appendix A), was significantly upregulated in the Δ*Fgmdm33-3* and Δ*Fgmdm33-5* mutants relative to that with PH-1 and cΔ*Fgmdm33* (Figure 5D). These results indicated that *FgMDM33* may negatively regulate the apoptosis in *F. graminearum*.

## 4. Discussion

We have described that FgMdm33 was required for the maintenance of proper mitochondrial morphology in *F. graminearum*. This role of FgMdm33 was determined through the analysis of the mitochondrial structure in Δ*Fgmdm33*-mutant cells, which displayed mitochondrial aggregation and extension as elongated tubules (Figure 3). In *S. cerevisiae*, the Δ*mdm33* mutant bearing defects in the mitochondrial fission did not acquire a respiratory-deficient growth phenotype [17]. In *F. graminearum*, the Δ*Fgmdm33* mutants could grow on CM plates with lactate as the sole carbon source (Figure 3A), displaying a similar phenotype to the Δ*mdm33* mutant of *S. cerevisiae*, suggesting that the Δ*Fgmdm33* mutants may bear the mitochondrial fission defect. Moreover, the elongated mitochondria in Δ*Fgmdm33* mutants provided further evidence to confirm the role of *FgMDM33* in mitochondrial fission.

Mitochondria with an abnormal morphology can be degraded by mitophagy [42]; therefore, we speculated that *FgMDM33* regulates mitochondrial morphology by mitophagy. Unexpectedly, under the conditions of the induction of mitophagy, autophagic bodies in the vacuoles were observed in the Δ*Fgmdm33-5* mutant (Figure 4A). These results suggested that *FgMDM33* may not be involved in the mitophagy in *F. graminearum*. However, the Δ*Fgmdm33* mutant is defective in the proteolysis of GFP-FgAtg8 (Figure 4B), indicating that *FgMDM33* is required for non-selective macroautophagy. In *F. geaminearum*, several mitochondrial genes have been reported to be involved in autophagy. The deletion of *Fgporin*, a homologous gene of the yeast mitochondrial porin, located in the outer membrane, induced morphological changes and the dysfunction of mitochondria, which resulted in impaired autophagy [31]. The deletion mutant of mitochondrial *FgEchi* encoding enoyl-CoA hydratase exhibited increased autophagy [43]. These results support that the genes involved in mitochondrial homeostasis probably play important roles in the autophagy of *F. graminearum*.

Palmer et al. (2011) reported that proteins involved in mitochondrial morphology possibly play dual roles in both mitochondrial dynamics and apoptosis [44]. In this study, we found that the mitochondrial distribution and morphology family 33 gene *FgMDM33* is involved in the apoptosis of *F. graminearum*. The Δ*Fgmdm33* mutants were found to be more sensitive to 0.05% H_2_O_2_ relative to the wild-type strain PH-1 and the complemented strain cΔ*Fgmdm33* (Figure 5A,B). The qRT-PCR analysis demonstrated that the increased sensitivity of the Δ*Fgmdm33* mutants to H_2_O_2_ was attributable to the activation of apoptotic cell death, rather than the deficiency of the expression of catalase genes (Figure 5C,D). Emerging evidence suggests that mitochondria fission is closely associated with apoptosis [45]. Mitochondrial fission mediated by Drp1 (GTPase dynamin-related protein 1), the major driver of mitochondrial fission, contributes to baicalein-induced apoptosis and autophagy via the activation of the AMPK-signaling pathway [46,47]. The mitochondrial fission protein Fis1 can convey an apoptosis signal from the mitochondria to the ER [48]. In this study, we preliminarily elucidated that *FgMDM33* is involved in the apoptosis of *F. graminearum*. However, the roles of *FgMDM33* in apoptosis warrants further demonstration by the examination of the apoptotic process in Δ*Fgmdm33* mutants.

In *S. cerevisiae*, Mdm33 interacts with prohibitins, Phb1 and Phb2, which are the key components of the mitochondrial inner membrane homeostasis [21]. In addition, ERMES tethers the mitochondria to the ER and acts as a marker of the sites of mitochondrial fission, which maintains multiple interactions with the mitochondria [49,50]. Therefore, we performed yeast two-hybrid (Y2H) analyses to confirm the interaction between FgMdm33 and prohibitins, and the ERMES complex consisting of Mmm1, Mdm10, Mdm12, and Mdm34. Unfortunately, the interactions among FgMdm33 and FgMmm1, FgMdm10, FgMdm12, FgMdm34, FgPhb1, and FgPhb2 were, respectively, negative (Appendix A), implying that FgMdm33 may not interact with prohibitins and the ERMES complex in *F. graminearum*. Considering the important role of FgMdm33 in autophagy, we speculated that FgMdm33 may interact with autophagy-related proteins. In the future, large-scale screening assays by Y2H analyses need to be conducted to determine whether FgMdm33 interacts with autophagy-related proteins in *F. graminearum*.

## Figures and Tables

**Figure 1 jof-10-00579-f001:**
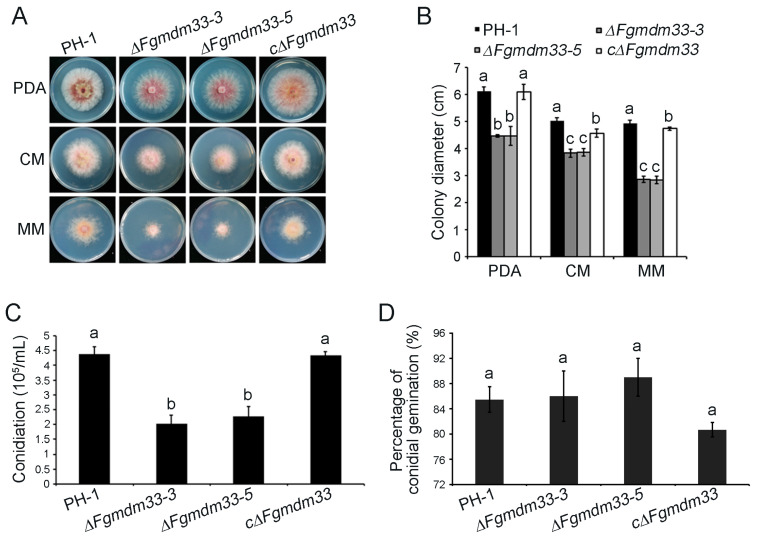
Δ*Fgmdm33* mutants are defective in mycelial growth and conidiation. (**A**), Colony morphology of the wild-type strain PH-1, the Δ*Fgmdm33* mutants, and the complemented strain cΔ*Fgmdm33* grown on PDA, CM, and MM plates at 25 °C for 3 days. (**B**), The colony growth of each strain in A. Error bars represent the standard deviation. Different small letters indicate a significant difference (*p* < 0.05). (**C**), A bar chart depicting the statistical analysis of conidiation. Conidia produced by each strain were quantified with a hemacytometer after incubation in the CMC liquid medium for 4 days in the light. Different small letters indicate a significant difference (*p* < 0.05). (**D**), The percentage of conidial germination. The conidia of each strain were cultured in a 2% sucrose solution and examined under a light microscope after 24 h. The same small letters indicate no significant difference (*p* < 0.05).

**Figure 2 jof-10-00579-f002:**
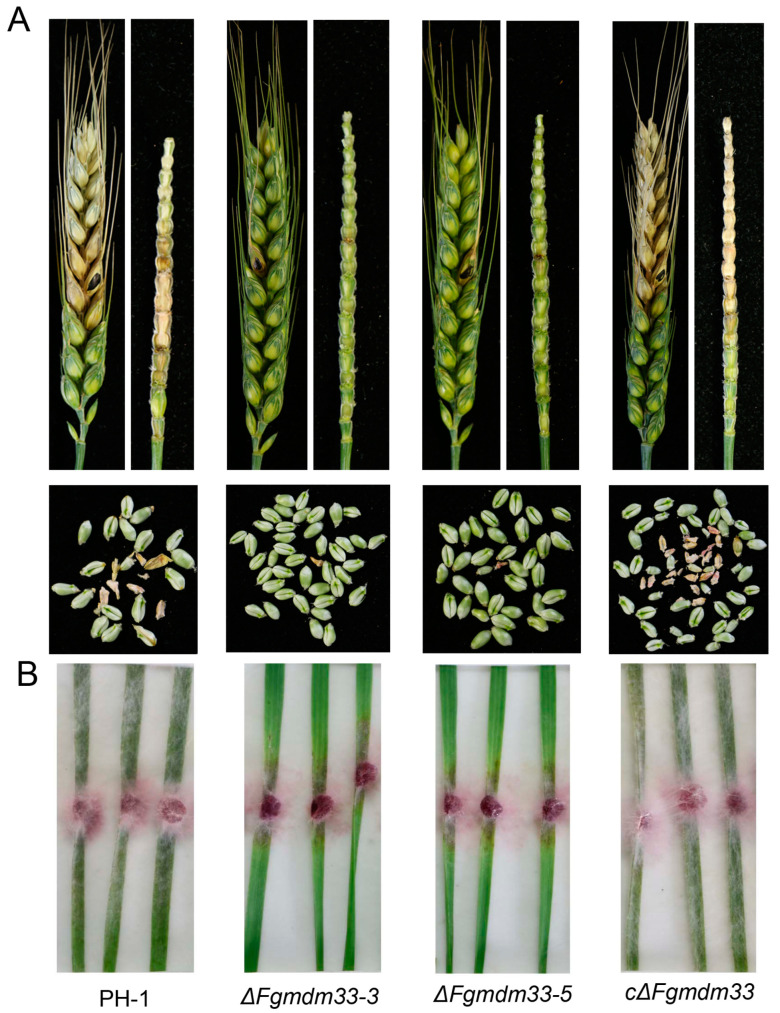
*FgMDM33* is required for full virulence. (**A**), The dissection of wheat heads infected by the wild-type strain PH-1, the Δ*Fgmdm33* mutants, and the complemented strain cΔ*Fgmdm33*. Inoculated wheat heads were examined and dissected after 14 days of inoculation. (**B**), Wheat cut-leaf assays. The mycelial plugs of each strain were inoculated onto the detached wheat leaves. The photographs were taken after 3 days of inoculation.

**Figure 3 jof-10-00579-f003:**
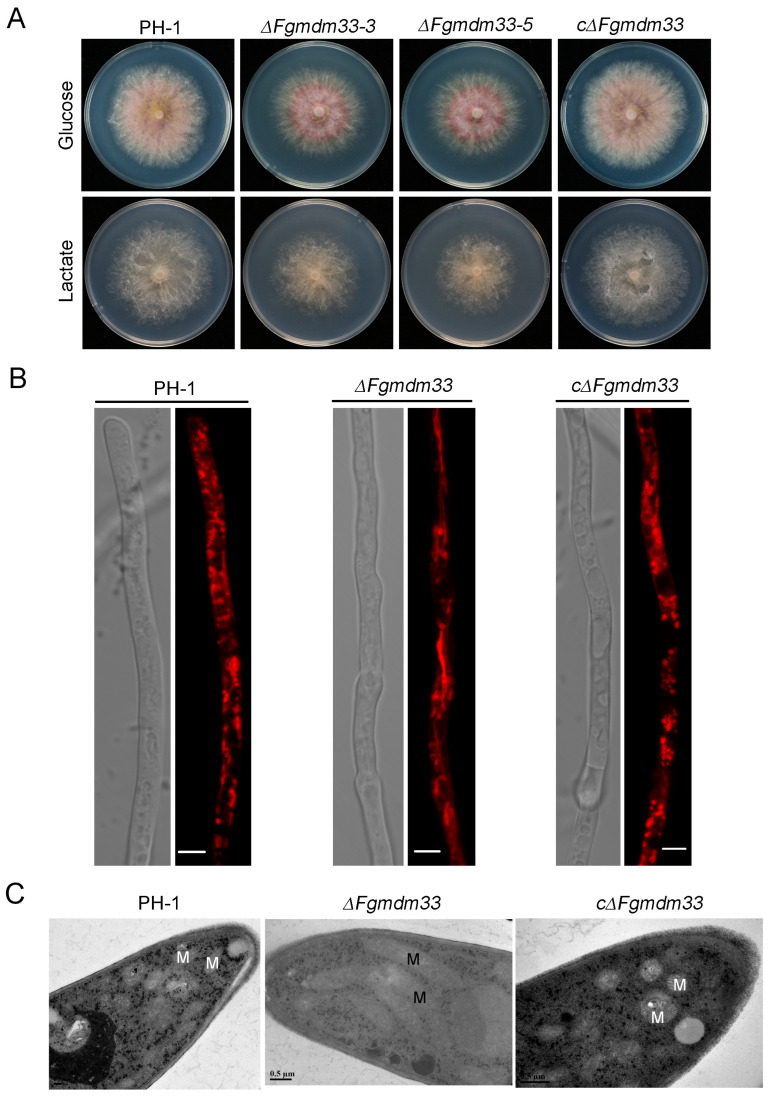
*FgMDM33* may be involved in mitochondrial fission. (**A**), Colony growth of the wild-type strain PH-1, the Δ*Fgmdm33* mutants, and the complemented strain cΔ*Fgmdm33* on CM plates containing glucose or lactate. (**B**), The mitochondria in the hyphal cells of the Δ*Fgmdm33* mutant were elongated. Each strain was grown in a CM liquid medium for 24 h at 25 °C, after which the hyphae were harvested and stained with MitoTracker^TM^ Red for microscopic observation. Scale bar = 5 μm. (**C**), Ultrastructural morphology of the mitochondria in the hyphal cells of each strain was visualized by transmission electron microscopy. M, mitochondria. Scale bar = 0.5 μm.

**Figure 4 jof-10-00579-f004:**
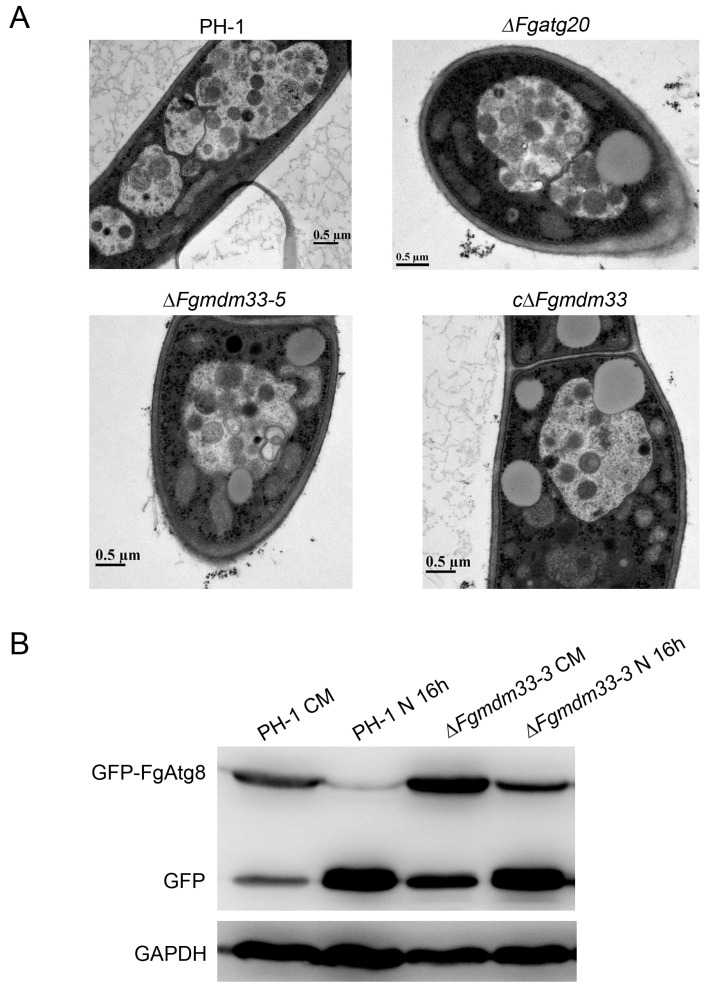
The deletion of *FgMDM33* caused defects in non-selective macroautophagy. (**A**) *FgMDM33* is dispensable for mitophagy in *F. graminearum*. The observation of autophagic bodies in the vacuoles of wild-type strain PH-1, the Δ*Fgmdm33* mutants, and the complemented strain cΔ*Fgmdm33*. Each strain was cultured in liquid CM at 25 °C for 24 h, after which it was shifted to BM-G for 30 h and starved in MM-N for another 24 h. Scale bar = 0.5 μm. (**B**) GFP-FgAtg8 proteolysis assays. Immunoblot assays with the total lysates from CM- or MM-N-cultured strains with an anti-GFP antibody or anti-GAPDH antibody.

**Figure 5 jof-10-00579-f005:**
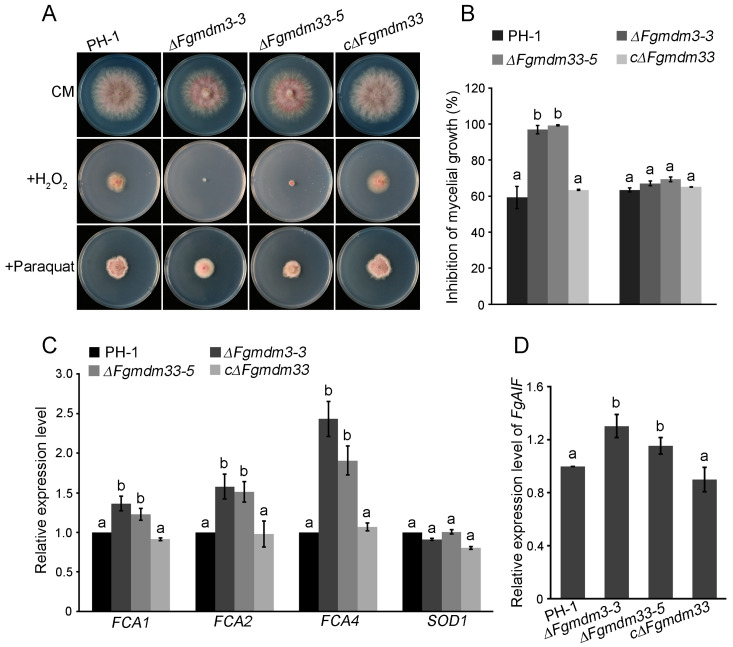
*FgMDM33* may be involved in apoptosis. (**A**) Colonies of the wild-type strain PH-1, the Δ*Fgmdm33* mutants, and the complemented strain cΔ*Fgmdm33* on CM with 0.05% H_2_O_2_ or 200 ppm paraquat for 3 days at 25 °C. (**B**) The statistical analysis of the percentage of mycelial growth inhibition. Line bars in each column represent the standard deviation. Different small letters indicate a significant difference (*p* < 0.05). (**C**) The transcriptional levels of *FgFCA1*, *FgFCA2*, *FgFCA4*, and *FgSOD1* by qRT-PCR. Line bars in each column represent the standard deviation. Different small letters indicate a significant difference (*p* < 0.05). (**D**) The expression of *FgAIF1* was upregulated in the Δ*Fgmdm33* mutants. Line bars in each column represent the standard deviation. Different small letters indicate significant differences (*p* < 0.05).

## Data Availability

The original contributions presented in the study are included in the article; further inquiries can be directed to the corresponding authors.

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
