# Peer review of "The Mitochondrial Distribution and Morphology Family 33 Gene FgMDM33 Is Involved in Autophagy and Pathogenesis in Fusarium graminearum"

_jof, 2024, doi:10.3390/jof10080579_

Round 1
Reviewer 1 Report
The manuscript presents a complete characterisation of the mdm33 gene in Fusarium graminearum (Fg).
However, the different approaches are not related and dont respond to a clear biological question or working hypothesis of the role of mdm33 in Fg
Please review the comments above. The characterisation of mdm33 is nicely detailed but the purpose to conduct such analysis is missing to my understanding
e.g. why the autophagy, mitochondrial function etc..., in Fg are important in the Fg in the pathosystem context?
Please answer this question with your results and formulate the hypothesis accordingly
Author Response
We feel great thanks for your professional review work on our article. These comments are valuable and helpful for improving our article.
Comments 1: No background of the importance of Fg, virulence mechanisms and alternatives to control no clear working hypothesis formulated, three or four studies on the mutant strain were mentioned but without any connection.
Response 1: Thanks for your valuable suggestion. We have revised the Introduction section. (Pages 4-5, Lines 76-87)
Comments 2:
Please review the comments above. The characterisation of mdm33 is nicely detailed but the purpose to conduct such analysis is missing to my understanding
e.g. why the autophagy, mitochondrial function etc..., in Fg are important in the Fg in the pathosystem context?
Please answer this question with your results and formulate the hypothesis accordingly.
Response 2: Thanks for your suggestion. Because the mitochondria play a central role in the cellular metabolism, mitochondrial function is important in F. graminearum. In this study, the results showed that FgMDM33 is involved in the mitochondrial fission, which is important for the mitochondria homeostasis. When this gene was knocked out, the mitochondrial morphology was affected and the vegetative growth of F. graminearum was severely defective. Mitochondria with abnormal morphology can be degraded by autophagy, and the results showed that the loss of FgMDM33 caused partial defects in autophagy. Therefore, we speculate that loss of FgMDM33 causes abnormal mitochondrial morphology, and then results in defects in cellular metabolism, and eventually plays important roles in F. geaminearum.
Thanks again. We have tried to answer this question. If this answer does not meet your criteria, please forgive us and contact us again.
Reviewer 2 Report
Dear Colleagues. Overall the work is interesting.
There are several questions about the methods
Line 100.
What type of complete medium did you use?
Line 102.
Why did you use CMC medium and not wash off the conidia from the agar surface? Have you looked at the germination of micro or macroconidia?
Line 103.
Why was sucrose needed? Fusarium conidia germinate in water.
Line 117-119.
Peroxide participates in redox reactions and is quickly destroyed in the presence of metal ions. Did she interact with the environment? This could be tested with the iodine-starch reaction. What was the mechanism of action of paraquat? To avoid artifacts, it would be possible to test the effect of oxidative stress on the germination of conidia in water.
Line 122. Why were these carbon sources chosen?
Could protein Mdm33 be a target for potential fungicides?
Author Response
We feel great thanks for your professional review work on our article. These comments are valuable and helpful for improving our article.
Comments 1: Line 100.
What type of complete medium did you use?
Response 1: Thanks for careful check. We have added the formula of complete medium. (Page 6, Lines 1113-114)
Comments 2: Line 102.
Why did you use CMC medium and not wash off the conidia from the agar surface? Have you looked at the germination of micro or macroconidia?
Response 2: Fusarium graminearum stain PH-1 used in this study cannot produce conidia on PDA plates, so we used CMC liquid medium to facilitate the production of conidia.
Thanks for your valuable suggestion. Because we did not observed micro or macroconidia in our lab, the germination of micro or macroconidia was not performed.
Comments 3: Line 103.
Why was sucrose needed? Fusarium conidia germinate in water.
Response 3: Thanks for your suggestion. 2% sucrose solution can promote the conidial germination better.
Comments 4: Line 117-119.
Peroxide participates in redox reactions and is quickly destroyed in the presence of metal ions. Did she interact with the environment? This could be tested with the iodine-starch reaction. What was the mechanism of action of paraquat? To avoid artifacts, it would be possible to test the effect of oxidative stress on the germination of conidia in water.
Response 4: Thanks for your valuable suggestion. We agree that more tests would be useful to understand the details of oxidative stress. Due to the change of workplace, we do not have the necessary tool-set to perform the iodine-starch reaction and test the effect of oxidative stress on the germination of conidia in water. Please kindly understand. We believe that the current results can shed some light on the sensitivity of mycelial growth to oxidative stress on medium plates. Thanks again.
Comments 5: Line 122. Why were these carbon sources chosen?
Could protein Mdm33 be a target for potential fungicides?
Response 5: The glucose was generally used as the a fermentative carbon source. The phenotype of the yeast Δmdm33 mutant displays no mtDNA loss and no acquisition of a respiratory-deficient growth. Therefore, the lactate as a non-fermentative carbon source was used. And, thanks for your suggestion. Mdm33 may be a target for potential fungicides, and this studies provided some scientific basis.
Round 2
Reviewer 2 Report
Dear colleagues. Thank you for your reply, I wish you good luck in your further research.
Sorry, in the corrected version I did not find the pictures that were in the previous one

Author Response
Comments 1: Sorry, in the corrected version I did not find the pictures that were in the previous one.
Response 1: We are sorry about this. The corrected version in this system only presented the revised manuscript because the pictures in the previous version have not been revised. We have uploaded the pictures in the attachment to this answer.
